# C-reactive protein: An easy marker for early differentiation between leptospirosis and dengue fever in endemic area

Olivier Maillard [1,2]*, David Hirschinger[3], Samuel Bénéteau[1], Yatrika Koumar[4], Adrien Vague[5], Rémi Girerd[5], Laura DiAscia[3], Julien Jabot[6], Julien Cousty[7], Andry Randrianjohany[8], Antoine Bertolotti[2,4], Loïc Raffray[9,10]

1 Department of Public Health and Research, CHU Réunion, Saint-Pierre, Reunion, France, 2 Clinical Investigation Center, INSERM CIC 1410, CHU Réunion, Saint-Pierre, Reunion, France, 3 Department of Emergency Medicine, CHU Réunion, Saint-Denis, Reunion, France, 4 Department of Infectious Diseases, CHU Réunion, Saint-Pierre, Reunion, France, 5 Department of Emergency Medicine, CHU Réunion, Saint-Pierre, Reunion, France, 6 Intensive Care Unit, CHU Réunion, Saint-Denis, Reunion, France, 7 Intensive Care Unit, CHU Réunion, Saint-Pierre, Reunion, France, 8 Department of Internal Medicine, Groupe Hospitalier Est Réunion, Saint-Benoit, Reunion, France, 9 Department of Internal Medicine, CHU Réunion, Saint-Denis, Reunion, France, 10 UMR Processus Infectieux en Milieu Insulaire Tropical (PIMIT), CNRS 9192, INSERM U1187, IRD 249, Université de La Réunion, Sainte-Clotilde, Reunion, France

☯ These authors contributed equally to this work.
* olmaillard@yahoo.fr

**Data Availability Statement:** All relevant data are within the paper and its Supporting Information files. The data underlying the results presented in

## Abstract

In tropical regions, leptospirosis and dengue fever (DF) are infectious diseases of epidemiological importance and have overlapping symptomatic features. The objective of this study was to identify the factors associated to diagnosing leptospirosis that differentiate it to DF at the initial hospital evaluation. A multicenter retrospective study was conducted comparing confirmed leptospirosis to DF cases. Clinical/laboratory findings were compiled at hospital admission on Reunion Island between 2018 and 2019. Multivariable logistic regression was used to identify the predictors of leptospirosis. In total, 98 leptospirosis and 673 DF patients were included with a mean age of 47.8 (±17.1) and 48.9 (±23.3) years, respectively. In the multivariate analyses, the main parameters associated with leptospirosis were: i) increased neutrophil counts, ii) C-reactive protein values, iii) the absence of prolonged partial thromboplastin time, and iv) a decrease of platelets. The most discriminating parameter was C-reactive protein (CRP). With a threshold of 50mg/L, CRP taken alone had a sensitivity of 94% and a specificity of 93.5%. The positive and negative likelihood ratios were 14.5 and 0.06, respectively. In the setting of an early presumptive diagnosis, we found that an increased CRP value (>50 mg/L) could help diagnose leptospirosis and aid the decision process for hospital surveillance and/or a potential antibiotic treatment regimen.

## Introduction

Leptospirosis is a bacterial zoonosis that exists throughout the world. However, it is more predominant in the tropical zones because of climatic and socio-demographic conditions being

the study are available from the INSERM CIC 1410
(cic@chu-reunion.fr).

**Funding:** The author(s) received no specific
funding for this work.

**Competing interests:** The authors have declared
that no competing interests exist.

more favorable to its propagation. The disease has become more prevalent due to climate
change and it has a high morbi-mortality rate with an estimate of more than 1 million cases
each year and a 6% mortality rate [1, 2]. Similarly, dengue fever (DF) is spreading throughout
the world as a consequence of the proliferation of the *Aedes* mosquitos. Its spread has been
intensified by several conditions such as uncontrolled urbanization, demographic changes,
increasing mobility and climate change [3, 4]. Epidemics of DF will probably be more frequent
in the future since it has been the case on Reunion Island where major seasonal outbreaks
since late 2017 have occurred [5].

During these epidemics, many concerns have been raised about the risk of confusion
between leptospirosis and DF because the clinical-biological presentation during the acute
phase of both diseases are often similar [6–11]. This confusion can be detrimental and carry a
risk of delayed antibiotic treatment leading to increased mortality in leptospirosis patients
[12–14], particularly when leptospirosis is misdiagnosed as DF [10, 15].

A point-of-care diagnosis would be needed in primary care. During the first days of illness,
diagnosis of leptospirosis or DF using PCR techniques are very efficient, albeit costly. Serological
testing is cheaper but has less sensitivity. Confirmation using these techniques is often
delayed because laboratories can be quickly overwhelmed during major outbreaks and rapid
diagnostic tests (RDT) could be used instead but are not always available and may also have
diagnosis accuracy discrepancies depending on their setting [7, 8]. The objective of this study
was to identify the clinical and biological factors associated to diagnosing leptospirosis that dif-
ferentiate it to the detection of DF at hospital admission on the Reunion Island where the two
diseases are endemic and epidemic.

## Methods

### Study design and patient selection

We conducted a comparative retrospective study with one group of leptospirosis patients that
were selected and compared to a group of DF patients. Leptospirosis and DF cases were
recruited at the four main hospitals on Reunion Island. Eligible patients were screened by the
microbiology laboratory at the University Hospital of Reunion where all PCR and serological
tests were centralized. Patients' characteristics were compiled from electronic medical records
of the study hospitals. Since early diagnosis is crucial to alleviating leptospirosis severity and
hospital monitoring, we focused on the clinical and basic laboratory test results at the time of
hospital admission.

Consecutive confirmed leptospirosis cases with clinical information in their medical files
and at least one biological sample were retrospectively identified between January 1, 2018 and
September 11, 2019 and included in the study. Consecutive DF patients were recruited from
the EPIDENGUE retrospective database (National Health Data Hub registration number:
F20201021104344) providing that they also had a clinical evaluation and at least one biological
sample at admission and were screened between January 1, 2019 and June 6, 2019 which corre-
sponded with the 2019 seasonal epidemic-level peak of dengue.

### Confirmation of cases

In regard to leptospirosis, a confirmed case was defined by either a positive PCR blood test,
urine sample or positive serological test for IgM. Suspected DF cases were screened by either a
positive PCR blood test or by a positive rapid strip test for an NS1 antigen or IgM serology.
Due to the diagnosis accuracy discrepancies of serology and rapid tests, we only selected den-
gue cases that were confirmed by PCR for analyses.

Confirmation tests were performed at the microbiology laboratory of University Hospital Center of Reunion Island. For the PCR technique, a test combining a search for *Leptospira*, dengue fever and chikungunya virus genomes was performed in a single-step, real-time RT-PCR with TaqMan probes as previously described in Giry et al (2017) [16]. For *Leptospira* serology, the Serion ELISA classic IgM (Institut Virion\Serion GmbH, Germany) was used. Interpretation of results was done according the manufacturer's instructions: anti-leptospiral IgM <15 IU/ml gave a negative result suggesting no evidence of a recent infection, 15–19 IU/ml gave a borderline result suggesting that it may be a recent infection and ≥20 IU/ml gave a positive result which was interpreted as a recent or current infection to be compared with the result of a second IgM serology often performed in the hospital, because leptospirosis cases are usually re-examined 1 month after diagnosis.

## Data collection and definitions

For all patients, we collected demographic data as well as the date of symptom onset and the date of the first visit to the hospital, which was usually to the emergency room. Clinical features at admission were reported for all patients using medical chart review examination. Symptoms and physical signs not reported on medical charts were considered absent. Biological indicators were reported using the blood sample performed upon admission at the same time as the clinical observations.

## Statistical methods

A bivariate analysis was first performed to describe and compare the two groups of patients. Continuous variables normality was checked by the Shapiro-Wilk test and the homogeneity of variances by the Levene test. These variables were described with means and standard deviations and differences were tested using the Student's t-test or the Wilcoxon-Mann-Whitney test, as appropriate. Categorical variables were described as numbers and percentages. They were subsequently compared using the Chi square test or Fisher's exact test.

A multivariate logistic regression model was performed using variables with a p<0.20 in bivariate analysis. Crude and adjusted odds ratios (OR) were computed with 95% confidence intervals (95% CI). A backward-stepwise process was used to select the final model. Goodness-of-fit was assessed with the Hosmer and Lemeshow test. Receiver operating characteristic (ROC) curves and AUC (area under the curve) were performed to compare the final model to a parsimonious model using the DeLong test while also assessing the best thresholds to maximize the Youden's index. Performance criteria (sensitivity, specificity and positive and negative likelihood ratios) were assessed for the best model to accurately distinguish leptospirosis from DF. Predictive values and post-hoc probabilities were also estimated in the condition of the sampling. All tests were two-tailed and the significance level was set at 0.05. All analyses were carried out with SPSS (IBM SPSS 23.0, IBM Corp. Armonk, NY, USA) and R (R4.1.2, R Foundation for Statistical Computing, Vienna, Austria).

## Ethical statement

Informed consent was obtained from participants and data were anonymized. In accordance with French regulations, this observational study on previously collected data did not require any Ethics Committee approval but was registered with the National Health Data Hub under the number I04060704202020 (LEPTODENG2019 project). The study was conducted according to the Declaration of Helsinki and reference methodology MR-004 of the National Commission for Information Technology and Liberties (CNIL authorization 2206739) in compliance with General Data Protection Regulations (GDPR).

## Results

In total, 98 patients diagnosed with leptospirosis and 673 PCR-confirmed DF sufferers were included in our study (Fig 1). The main diagnosis tool was the PCR technique for leptospirosis (Fig 1). The mean age in the leptospirosis cases was 47.8 years (±16.9) and 48.9 years (±23.3) for DF cases. The gender ratio M:F was 10.1 for leptospirosis and 0.80 for DF ($p<0.001$).

All patients were enrolled at the time of their first evaluation at the hospital, which was usually in the emergency room. At hospital evaluation, the delay from symptoms onset was shorter in the DF group: 2.5 (±2.5) vs. 4 days (±2.1) (Table 1).

### Clinical and laboratory characteristics to distinguish leptospirosis from dengue

In the bivariate analysis, the clinical factors that were more significantly associated to leptospirosis ($p<0.001$) were asthenia, anorexia, muscle pain, cough, jaundice, hemoptysis and oliguria (Table 1). On admission, the values of the following parameters were significantly higher in leptospirosis patients compared to DF sufferers: leukocyte count, neutrophil count, monocyte count, creatinine, urea, bilirubin, creatine kinase and CRP (Table 2). Moreover, hematocrit, platelet count, potassium, sodium, chlorine and calcium levels were lower in the leptospirosis cases compared to the DF cases. Biological factors more frequently found in DF were: prolonged thromboplastin time, leukopenia and particularly, neutropenia inferior to 1.5 $\times10^9$/L with 17% of the patients compared to only 1% in leptospirosis ($p<0.001$). The number of missing biological values are indicated in S1 Table and main outcomes on the evolution/follow-up of confirmed cases are reported in S2 Table.

### Multivariate predictive model for leptospirosis diagnosis

Four biological parameters were retained in the final model as presented in Table 3. CRP had better adjusted Wald test value whereas other parameters taken alone had a lower contribution to the model.

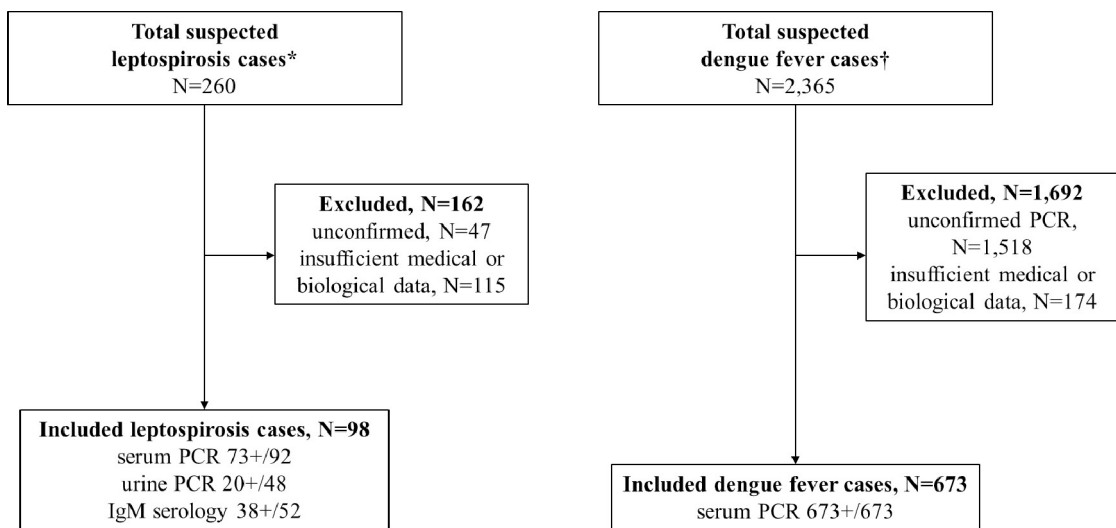

*Retrospective analysis of medical and laboratory charts between 2018 and 2019,
†Medical chart analysis of patients arriving at the hospital with febrile illness from 01/01/2019 to 06/30/2019

**Fig 1. Study flowchart and diagnostic methods.** *Retrospective analysis of medical and laboratory charts between 2018 and 2019, †Medical chart analysis of patients arriving at the hospital with febrile illness from 01/01/2019 to 06/30/2019.

**Table 1. Comparison of the main demographic and clinical factors at initial hospital evaluation of leptospirosis and dengue fever cases on Reunion Island from 2018 to 2019 (bivariate analysis).**

| Variable | Leptospirosis (N = 98) | Dengue fever (N = 673) | P value |
|---|---|---|---|
| Age, years, mean(±SD) | 47.8 (±17.1) | 48.9 (±23.3) | 0.519 |
| Male, n (%) | 89 (91) | 299 (44) | **<0.001** |
| Time before admission, days, mean(±SD) | 3.9 (±2.1) | 2.5 (±2.5) | **<0.001** |
| **Presenting symptoms** | | | |
| Asthenia, n (%) | 90 (92) | 482 (72) | **<0.001** |
| Anorexia, n (%) | 64 (65) | 324 (48) | **0.001** |
| Muscle pain, n (%) | 81 (83) | 375 (56) | **<0.001** |
| Joint pain, n (%) | 35 (36) | 279 (42) | 0.277 |
| Headache, n (%) | 42 (43) | 347 (52) | 0.107 |
| Mental confusion, n (%) | 2 (2) | 45 (7) | **0.042** |
| Faintness, n (%) | 1 (1) | 48 (7) | **0.005** |
| Shortness of breath, n (%) | 6 (6) | 74 (11) | 0.116 |
| Cough, n (%) | 18 (18) | 46 (7) | **0.001** |
| Chest pain, n (%) | 6 (6) | 28 (4) | 0.425 |
| Abdominal pain, n (%) | 38 (39) | 186 (28) | **0.027** |
| Diarrhea, n (%) | 28 (29) | 134 (20) | 0.057 |
| Vomiting, n (%) | 29 (30) | 182 (27) | 0.600 |
| Pruritus, n (%) | 3 (3) | 51 (8) | 0.071 |
| **Physical examination** | | | |
| Fever >38.5˚C, n (%) | 25 (26) | 242 (38) | **0.022** |
| Pulse rate, per min, mean(±SD) | 101 (±19) | 96 (±37) | 0.186 |
| DBP, mmHg, mean(±SD) | 70 (±14) | 73 (±13) | **0.024** |
| SBP, mmHg, mean(±SD) | 122 (±22) | 127 (±23) | **0.030** |
| Skin rash, n (%) | 2 (2) | 46 (7) | **0.037** |
| Jaundice, n (%) | 36 (37) | 5 (0.7) | **<0.001** |
| Polyadenopathy, n (%) | 5 (5) | 7 (1) | **0.011** |
| Urine output <500ml*, n (%) | 24 (25) | 8 (1) | **<0.001** |
| Purpura, n (%) | 5 (5) | 34 (5) | 1 |
| Hemoptysis, n (%) | 10 (10) | 6 (0.9) | **<0.001** |
| Hematuria, n (%) | 10 (10) | 19 (3) | **0.002** |
| Other hemorrhagic symptoms, n (%) | 3 (3) | 76 (11) | **0.004** |

*During the first 24 hours of hospital stay

DBP: diastolic blood pressure; SBP: systolic blood pressure; SD: standard deviation

Overall, the final model resulted in a good fit-of-goodness (Hosmer and Lemeshow test, p = 0.587) with 37% of data missing, predominantly the aPTT ratio data. The results of a CRP test at admittance were lacking in 5.2% of all the cases.

## Performances of CRP in diagnosis

Since the CRP was the most discriminating parameter, we studied its screening performance when used alone compared to the final model. ROC curves were performed for the final model and CRP alone (Fig 2). Area under curves and the 95% CI were excellent for discriminating leptospirosis from dengue fever and no significant statistical difference was found between the AUC for the final model and the CRP alone model (DeLong test, p = 0.146).

According to the ROC curve of CRP taken alone, the best CRP cut-off value was 50mg/L. With this threshold, a diagnosis of leptospirosis was associated to an OR of 221 (95% CI: 91 to

**Table 2. Comparison of the main biological factors at initial hospital evaluation for leptospirosis and dengue fever cases on Reunion Island from 2018 to 2019 (bivariate analysis).**

| Variable, mean(±SD) | Leptospirosis (N = 98) | Dengue fever (N = 673) | P value |
|---|---|---|---|
| Hemoglobin, g/dL | 13.3 (±1.8) | 13.4 (±2.0) | 0.581 |
| Hematocrit, % | 37.9 (±5.0) | 39.3 (±5.2) | **0.010** |
| RBC mean corpuscular volume, fL | 83 (±9) | 85 (±6) | 0.101 |
| Leukocytes count, $\times10^9$/L | 10.8 (±4.0) | 5.1 (±3.3) | **<0.001** |
| Neutrophils count, $\times10^9$/L | 9.3 (±3.8) | 3.5 (±2.6) | **<0.001** |
| Lymphocytes count, $\times10^9$/L | 0.80 (±0.49) | 0.86 (±1.44) | 0.679 |
| Monocytes count, $\times10^9$/L | 0.67 (±0.38) | 0.56 (±0.34) | **0.003** |
| Platelets count, $\times10^9$/L | 103 (±72) | 165 (±79) | **<0.001** |
| aPTT ratio over control value | 1.10 (±0.13) | 1.18 (±0.22) | **0.005** |
| Plasma creatinine, μmol/L | 278 (±268) | 106 (±108) | **<0.001** |
| Blood urea nitrogen, mmol/L | 13.8 (±11.1) | 5.6 (±4.7) | **<0.001** |
| AST, IU/L | 104 (±104) | 101 (±541) | 0.957 |
| ALT, IU/L | 69 (±56) | 60 (±225) | 0.707 |
| ALP, IU/L | 94 (±34) | 83 (±54) | 0.082 |
| GGT, IU/L | 86 (±73) | 65 (±144) | 0.230 |
| Total bilirubin, μmol/L | 84.5 (±103.6) | 10.9 (±33.7) | **<0.001** |
| Serum sodium, mmol/L | 133.3 (±4.4) | 136.5 (±3.3) | **<0.001** |
| Serum potassium, mmol/L | 3.7 (±0.5) | 3.9 (±0.5) | **<0.001** |
| Serum chlorine, mmol/L | 93.2 (±5.5) | 98.9 (±4.4) | **<0.001** |
| Serum calcium, mmol/L | 2.21 (±0.13) | 2.26 (±0.12) | **0.002** |
| CK, IU/L | 2144 (±3176) | 608 (±7145) | **<0.001** |
| CRP, mg/L | 229.2 (±122.1) | 18.5 (±30.3) | **<0.001** |

ALP: alkaline phosphatase; aPTT: activated partial thromboplastin time; AST: aspartate aminotransferase; ALT: alanine aminotransferase; CK: creatine kinase; CRP: C-reactive protein; GGT: gamma-glutamyl transferase; RBC: red blood cell; SD: standard deviation

536). For the diagnostic prediction of leptospirosis, the performances of CRP>50mg/L are detailed in Table 4.

Under sampling conditions, the post-test probability of having leptospirosis with CRP <50mg/L was 1% [CI 95% (0 to 2)].

Of the 98 leptospirosis patients, 6 presented in emergency room with CRP<50mg/L (false negatives), of whom 5 in the first two days of illness onset and one at Day 11. Forty-one DF patients (6%) were retrieved with CRP>50mg/L. They were older with more severe medical conditions. Most of them (85%) consulted the emergency room in the first three days of symptoms. Seventeen DF patients with CRP>50mg/L (41%) were coinfected and 16 cases without coinfection had other conditions associated with an elevated CRP at clinical examination.

**Table 3. Multivariate logistic regression analysis for the diagnosis of leptospirosis compared to dengue fever at initial hospital evaluation on Reunion Island from 2018 to 2019.**

| Variable | aOR | 95% CI | P-value | Wald test value |
|---|---|---|---|---|
| Polymorphonuclear neutrophil cells | 1.29 | 1.10 to 1.52 | 0.002 | 9.65 |
| Platelets | 0.98 | 0.97 to 0.99 | <0.001 | 16.43 |
| aPTT ratio | 0.001 | 0 to 0.027 | <0.001 | 15.06 |
| C-reactive protein | 1.03 | 1.02 to 1.04 | <0.001 | 33.09 |

aOR: adjusted odds ratio; aPTT ratio: ratio of activated partial thromboplastin time of patient over control

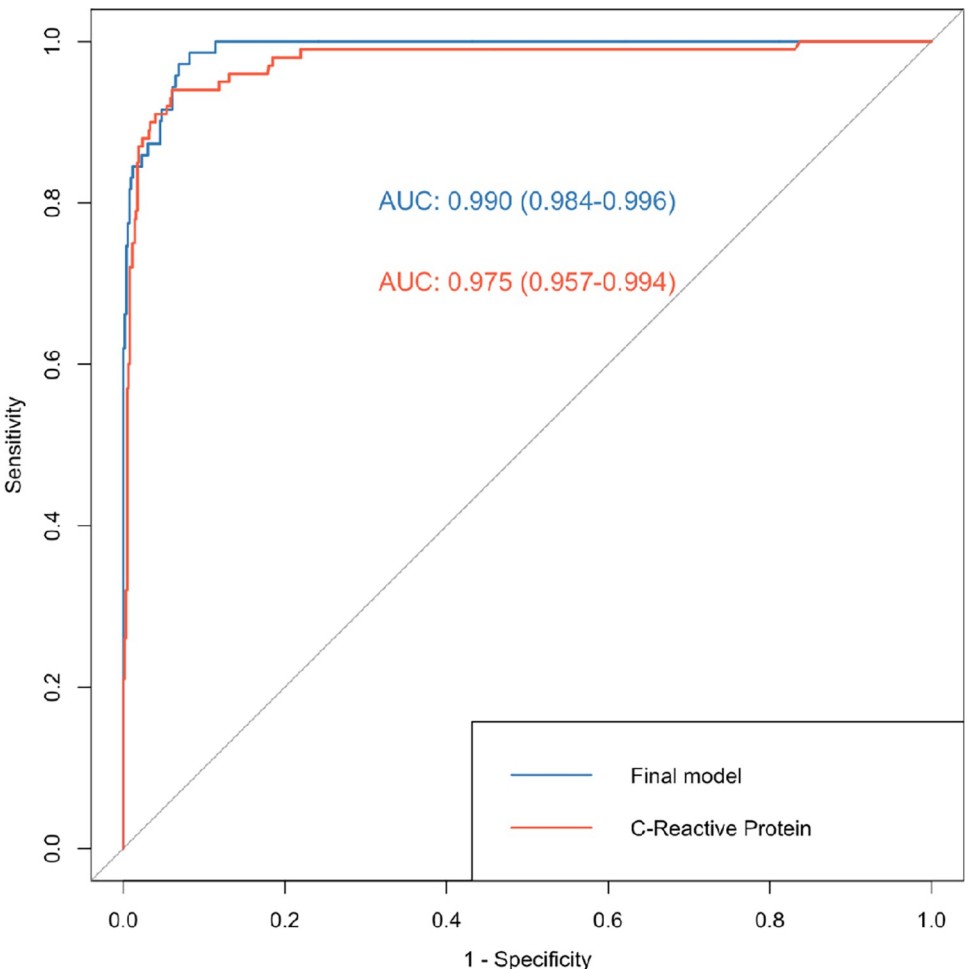

AUC; area under the curve with 95% confidence interval (95% CI) in parenthesis

**Fig 2. Receiver operating characteristic (ROC) curve in the diagnostic prediction of leptospirosis versus dengue fever according to the final model and with the C-reactive protein (CRP) alone.** AUC; area under the curve with 95% confidence interval (95% CI) in parenthesis.

**Table 4. Performance of the C-reactive protein (CRP) cut-off at 50mg/L in differentiating leptospirosis from dengue fever cases at hospital admission on Reunion Island from 2018 to 2019.**

| Statistic | Value | 95% CI |
|---|---|---|
| **Intrinsic properties** | | |
| Sensitivity | 0.939 | 0.872 to 0.977 |
| Specificity | 0.935 | 0.913 to 0.953 |
| Positive Likelihood Ratio | 14.49 | 10.73 to 19.57 |
| Negative Likelihood Ratio | 0.070 | 0.030 to 0.140 |
| **Extrinsic properties** | | |
| Positive predictive value | 0.692 | 0.624 to 0.752 |
| Negative predictive value | 0.990 | 0.979 to 0.995 |
| Accuracy | 0.936 | 0.915 to 0.952 |

CI: confidence interval

Comparison of characteristics between DF patients according to this CRP threshold (false positives vs true negatives) are provided in S3 Table.

## Discussion

Achieving accurate leptospirosis diagnosis during a DF outbreak is challenging because of overlapping symptoms and the associated risk of delayed antibiotic therapy. This comparative study shows that the use of CRP alone with a cut-off value of 50mg/L is an effective biomarker to differentiate leptospirosis from DF before laboratory confirmation is received. Our findings may be helpful for clinicians, particularly in tropical zones who need to facilitate timely antibiotic administration in leptospirosis cases as well as adjust hospital monitoring. It should also be considered for travelers returning from endemic/epidemic areas.

Our inclusion criteria used consensual biological confirmation for both leptospirosis and DF that allowed for a robust case definition. In addition, the most used confirmation test was a blood PCR technique with a unique procedure that systematically assessed both the presence of *Leptospira* and DF virus in the blood at the same time. Another strength which improved the representativeness of our study was the consecutive recruitment of cases that provided at least one initial available biological sample.

The activities at risk of potential leptospirosis infection exposure is a well-known item for leptospirosis diagnosis scores when facing an acute febrile illness in tropical zones [10, 17]. However, this data was largely unknown in DF patients and could not be analyzed due to the retrospective method of the study. Moreover, exposure information is mostly declarative and depended on the consciousness level and recollection of patients during the previous 14 to 21 days. This could have induced a memory bias. The gender ratio could also be a relevant factor but it cannot be used for screening even if it could ultimately reinforce a presumptive clinical diagnosis of leptospirosis. For this reason, these variables, albeit significant, were not entered into the final model to assess independent factors for an early diagnosis at hospital admission.

In the bivariate analysis, some clinical and demographical factors had significant association to either leptospirosis or DF. For instance, male gender, cough, muscle pain or jaundice were associated with leptospirosis. Surprisingly, some parameters were not retrieved such as skin rash or joint pain, although they are usually associated with DF. Other clinical characteristics have been described as indicative for clinicians to be able to differentiate one disease from the other [6, 10] but data between studies are contradictory [7, 9, 10] (S4 Table). Therefore, clinical features fail to serve as accurate discriminating factors [18]. It is not surprising that the main variables retained in the multivariate models of many studies are biological parameters [7, 9, 19, 20] which was also the case for our study.

As shown in previous studies, leukocyte count or, more specifically, neutrophil count are currently used as a critical element to differentiate leptospirosis from DF particularly because leptospirosis has a trend for neutrophilia whereas DF trends towards neutropenia which our study and literature have confirmed [7, 9, 10, 21]. Similarly, elevated bilirubin level was also shown to be linked to leptospirosis [7, 10]. However, in the context of emergency management, these two parameters (neutrophil count and bilirubin) have not matched the performance of CRP for presumptive diagnosis of leptospirosis because they add none or little useful information compared to CRP alone. It may also be worth noting that regarding the thromboplastin time, this parameter was not explored in most studies although sometimes was prolonged in DF patients [22].

According to our study and a previous retrospective study in French Guiana, high CRP seems to be a valuable biomarker in order to differentiate leptospirosis from DF [7]. Unfortunately, this parameter was not evaluated in other studies comparing the two diseases [9, 16,

17]. In Le Turnier at al., CRP used alone was almost as performant as the multi-parametric model used for the presumptive diagnosis of leptospirosis [7]. However, CRP is not specific and therefore it cannot be excluded that patients may have a high CRP level from another condition such as chronic illness or bacterial co-infection which can also occur during DF [23]. This was the case for 41 DF patients (6% of the total). Moreover, CRP has been used as a marker of DF severity but the cut-off values used were approximately 20 to 30 mg/L. [24, 25] which is clearly below the threshold of 50mg/L established in our work which targeted leptospirosis diagnosis and not a DF severity assessment. Overall, clinicians should keep in mind the need to rule out other febrile illnesses in cases of elevated CRP.

The main limitation of our study was the retrospective approach that could have weakened the data collection accuracy and statistical power since some variables could not be properly compared due to missing information. This model will need to be prospectively tested and validated in order to determine its potential clinical use. Since other infectious diseases may mimic their clinical presentation, namely bacterial sepsis, malaria, influenza, Covid-19 and rickettsial diseases, another limitation was the comparison of leptospirosis to DF only. Thus, our results should be interpreted with caution and not systematically transposed to areas/settings where such illnesses are largely prevalent.

## Conclusion

Leptospirosis can be difficult to diagnose and differentiate from DF, especially during epidemic periods. Since delaying the initiation of antibiotic therapy may be detrimental in leptospirosis cases, clinicians need a rapid and accurate diagnosis. For these reasons and regardless of the limitations mentioned above, CRP could be a rapid and cost-effective biomarker to differentiate leptospirosis from DF in tropical zones. CRP values may be helpful in deciding whether to initiate antibiotics while waiting for the results of more specific biological tests. It can also be helpful for clinicians to decide whether to hospitalize cases identified as at risk of leptospirosis or of DF with co-infections that could also worsen the course of these diseases and necessitate an antibiotic regimen.

## Supporting information

**S1 Table. Comparison of clinical and biological factors at initial hospital presentation of leptospirosis and dengue fever cases on Reunion Island between 2018 and 2019 (bivariate analysis).** *Regarding symptoms and other physical signs not reported, if a symptom/sign was not mentioned during the retrospective review of the medical charts then it was considered as absent. Therefore, there is no data considered as missing data for the symptom category. ALP: alkaline phosphatase; aPTT: activated partial thromboplastin time; AST: aspartate aminotransferase; ALT: alanine aminotransferase; BUN: blood urea nitrogen; CK: creatinine kinase; CRP: C-reactive protein; DBP: diastolic blood pressure; GGT: gamma-glutamyl transferase; RBC: red blood cell; SBP: systolic blood pressure.
(DOCX)

**S2 Table. Main outcomes of confirmed cases of leptospirosis and dengue fever on Reunion Island during 2018 and 2019.** ICU: intensive care unit.
(DOCX)

**S3 Table. Comparison of DF patients characteristics according to CRP threshold of 50mg/L on Reunion Island during 2018 and 2019.** ICU: intensive care unit.
(DOCX)

**S4 Table. Main studies reporting differentiating factors between leptospirosis and DF in multivariate analysis.** aPTT ratio: ratio of activated partial thromboplastin time of patient over control; AST: aspartate transaminase; CK: creatinine kinase; ESR: erythrocyte sedimentation rate; LR-: negative likelihood ratio, LR+; positive likelihood ratio, ND; not determined, MAT; microscopic agglutination test; NS1Ag: non-structural 1 antigen; PCR: polymerase chain reaction; Se: sensitivity; Sp: specificity; WBC: white blood cells. *Values are inferred from sensitivity and specificity from the studies according to the formulae: LR+ = Se/(1-Sp) and LR- = (1-Se)/Sp.
(DOCX)

## Acknowledgments

The authors would like to thank the staff from the clinical and biological departments who took part in patient care and study implementation as well as AcaciaTools for their reviewing services.

## Author Contributions

**Conceptualization:** Olivier Maillard, David Hirschinger, Antoine Bertolotti, Loïc Raffray.

**Data curation:** David Hirschinger, Yatrika Koumar, Adrien Vague, Rémi Girerd, Laura DiAscia, Julien Jabot, Julien Cousty, Andry Randrianjohany, Antoine Bertolotti, Loïc Raffray.

**Formal analysis:** Olivier Maillard, Samuel Bénéteau.

**Investigation:** David Hirschinger, Yatrika Koumar, Adrien Vague, Rémi Girerd, Laura DiAscia, Julien Jabot, Julien Cousty, Andry Randrianjohany, Antoine Bertolotti, Loïc Raffray.

**Methodology:** Olivier Maillard, David Hirschinger, Antoine Bertolotti, Loïc Raffray.

**Supervision:** Antoine Bertolotti, Loïc Raffray.

**Writing – original draft:** Olivier Maillard, Antoine Bertolotti, Loïc Raffray.

**Writing – review & editing:** David Hirschinger, Samuel Bénéteau, Adrien Vague, Rémi Girerd, Laura DiAscia, Julien Jabot, Julien Cousty, Andry Randrianjohany.

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
