## [Decision Letter · Decision Letter 0]

27 Feb 2023

PONE-D-22-26652C-reactive protein: an easy marker for early differentiation between leptospirosis and dengue fever in endemic areaPLOS ONE

Dear Dr. Maillard,

Thank you for submitting your manuscript to PLOS ONE. After careful consideration, we feel that it has merit but does not fully meet PLOS ONE’s publication criteria as it currently stands. Therefore, we invite you to submit a revised version of the manuscript that addresses the points raised during the review process.

Please address the comments by reviewer #2 related to the methodological limitations of the manuscript with respect to the diagnosis of leptospirosis and the generalizability of the results as other causes of fever were not included. However, as novelty is not required for publication in PLOS ONE, you do not need to address the comment. The CART analysis would be a welcome addition to the manuscript but is not necessary. 

We look forward to receiving your revised manuscript.

Kind regards,

Andrea L. Conroy, PhD

Academic Editor

PLOS ONE

Journal Requirements:

Furthermore, we noted that this is a retrospective study as such please could you clarify the nature of informed consent.

3. Please revise the ethics statement to clarify whether the data protection agency had specifically the approved the study described in the mansucript text.

Reviewers' comments:

Reviewer's Responses to Questions

**Comments to the Author**

1. Is the manuscript technically sound, and do the data support the conclusions?

Reviewer #1: Yes

Reviewer #2: Partly

2. Has the statistical analysis been performed appropriately and rigorously? 

Reviewer #1: Yes

Reviewer #2: Yes

3. Have the authors made all data underlying the findings in their manuscript fully available?

Reviewer #1: Yes

Reviewer #2: Yes

4. Is the manuscript presented in an intelligible fashion and written in standard English?

Reviewer #1: Yes

Reviewer #2: Yes

5. Review Comments to the Author

Reviewer #1: This is a scientific article that tends to demonstrate the interest of CRP determination in the diagnosis of leptospirosis versus dengue in the French department, Reunion Island. The article is of good quality, well written and very clear.

The conclusions are not completely new, and are largely in line with other similar articles, notably the one carried out in French Guyana. This, in my opinion, limits the interest of the article because there is no great novelty.

One of the major interests is the part aiming at studying the performance of the CRP by using ROC and AUC curves and which confirm the data.

The authors discuss the limitations of their studies: methodology (retrospective study) that may have limited the collection of data, particularly on clinical data. No or little epidemiological data. No special comments

Reviewer #2: General

Clear, well-written manuscript in fluent English

A major limitation is the study design being retrospective. Given the provided details, it is unclear to what extent a potential selection and/or information bias was introduced; no information on case-control matching is provided.

The distinct differences between gender distribution (matching?) and disease severity (severe vs non-severe dengue exhibits a broad CRP plasma level range) of the two disease groups, argue towards careful interpretation of the data, and preference of performing this study as a real life prospective evaluation.

No other comparator disease groups were included – i.e. no other diseases with similar clinical presentation characteristics like malaria, typhoid, rickettsial illnesses etc. were assessed for predictors – substantially limiting the interpretation of findings and cut-off selection for plasma CRP levels in a clinical setting.

The authors might reconsider expanding the study to include more and better-characterized samples, and perform a more sophisticated analysis including clinical and biochemical predictors and modelling to create a decision tree – this would result in an interesting publication.

Major comments

Analyses could have been expanded to include the construction of classification and regression trees, whereby using the additional clinic-laboratory findings to strengthen the predictions, versus CRP alone.

Minor comments

All dengue cases were PCR-confirmed, but Leptospirosis cases included a serological positivity endpoint (Serion ELISA, IgM), which has its well-described limitations. In addition, no positivity cutoff was provided, and no justification for cut off selection.

The authors state the limitations clearly at the end of the manuscript.

6. PLOS authors have the option to publish the peer review history of their article (what does this mean?). If published, this will include your full peer review and any attached files.

Reviewer #1: No

Reviewer #2: No

---

## [Author Response · Author response to Decision Letter 0]

11 Apr 2023

Dear Dr Andrea L. Conroy and Reviewers,

Please find attached our revised manuscript (PONE-D-22-26652), entitled ‘C-reactive protein: an easy marker for early differentiation between leptospirosis and dengue fever in endemic area’, by Olivier Maillard, David Hirschinger, Samuel Bénéteau, Yatrika Koumar, Adrien Vague, Rémi Girerd, Laura DiAscia, Julien Jabot, Julien Cousty, Andry Randrianjohany, Antoine Bertolotti, Loïc Raffray, which we are resubmitting for publication as an Original Article in PLOS One.

We want to sincerely thank all of you for your useful remarks that have helped us improve our paper and clarify its message.

Our responses to the comments are detailed below. Comments of the reviewers are in blue, responses are in black and recommended changes are in red in the attached files and in the manuscript.

REVIEWER 1

We wish to thank the reviewer for his helpful comments.

Comment: 

This is a scientific article that tends to demonstrate the interest of CRP determination in the diagnosis of leptospirosis versus dengue in the French department, Reunion Island. The article is of good quality, well written and very clear.

The conclusions are not completely new, and are largely in line with other similar articles, notably the one carried out in French Guyana. This, in my opinion, limits the interest of the article because there is no great novelty.

Response:

We agree with the reviewer that these are not new findings but with the exception of French Guiana, the performance and utility of CRP in differentiating leptospirosis from dengue fever has not been explored to date. In other settings, various other determinants have been reported. For this reason, the performance of CRP needed to be strengthened. Indeed, the symptoms and tropism of leptospirosis, thus the determinants, are very much related to the Leptospira species involved. Basing one’s decision on these determinants is context-dependent and not very useful elsewhere because the distribution of Leptospira species varies between settings. However, CRP appears to be a common determinant, regardless of the species or serogroup involved, and thus could be extended to all settings. A case-control study was conducted in French Guiana but we preferred to conduct a real-life study on consecutive confirmed cases of the 2 diseases to better assess the performance and utility of CRP at the point-of-care. Therefore, there was no matching in our comparative study. Although male gender and exposure history are associated with leptospirosis, it is not relevant to base one’s decision on these determinants.

In addition, we can also confirm the performance of CRP in Mayotte which has a very different distribution of Leptospira species compared to Reunion or French Guiana (personal data, publications in progress)

Comment: 

One of the major interests is the part aiming at studying the performance of the CRP by using ROC and AUC curves and which confirm the data.

The authors discuss the limitations of their studies: methodology (retrospective study) that may have limited the collection of data, particularly on clinical data. No or little epidemiological data. No special comments

Response: No response requested. We are pleased that the reviewer highlights one of the strengths and novelties of our study.

REVIEWER 2

We wish to thank the reviewer for his helpful comments.

Comment: 

Clear, well-written manuscript in fluent English

A major limitation is the study design being retrospective. Given the provided details, it is unclear to what extent a potential selection and/or information bias was introduced; no information on case-control matching is provided.

The distinct differences between gender distribution (matching?) and disease severity (severe vs non-severe dengue exhibits a broad CRP plasma level range) of the two disease groups, argue towards careful interpretation of the data, and preference of performing this study as a real life prospective evaluation.

Response:

We agree with the reviewer that a prospective study often brings more relevant results. Indeed we have already mentioned this limitation at the end of the discussion (page 18, lines 288-291):

“The main limitation of our study was the retrospective approach that could have weakened the data collection accuracy and statistical power since some variables could not be properly compared due to missing information. This model will need to be prospectively tested and validated in order to determine its potential clinical use.”

A case-control study was conducted in French Guiana but we preferred to conduct a real-life study on consecutive confirmed cases of the 2 diseases to better assess the performance and utility of CRP at the point-of-care. Therefore, there was no matching in our comparative study. Although male gender and exposure history are associated with leptospirosis, it is not relevant to base one’s decision on these determinants.

Therefore, the retrospective nature does not seem to bring as much bias since the diagnosis is based on biology and differentiation between the 2 diseases is mostly associated with biology rather than clinical data. During epidemics, multiplex PCR is routinely performed in patients presenting fever and CRP is a routine biological test in Reunion, which limits selection and information biases. Furthermore, prospective studies do not always succeed in including consecutive patients during epidemics, which increases the risk of selection bias. 

Furthermore, the symptoms and tropism of leptospirosis, thus the determinants, are very much related to the Leptospira species involved. Basing one’s decision on these determinants is context-dependent and not very useful elsewhere because the distribution of Leptospira species varies between settings. However, CRP appears to be a common determinant, regardless of the species or serogroup involved, and thus can be extended to all settings.

Finally, we can also confirm the performance of CRP in Mayotte which has a very different distribution of Leptospira species than Reunion or French Guiana (personal data, publications in progress)

Comment: 

No other comparator disease groups were included – i.e. no other diseases with similar clinical presentation characteristics like malaria, typhoid, rickettsial illnesses etc. were assessed for predictors – substantially limiting the interpretation of findings and cut-off selection for plasma CRP levels in a clinical setting.

Response:

Over the study period, only a few patients were diagnosed with any of these other diseases, mostly travellers, so we could not make comparison with these diseases in real-life during this period. So far, the incidence of these other diseases was very low compared to dengue or leptospirosis in our setting. It should be also noted that Reunion has been free of autochthonous malaria since 1974.

This limitation and the cautious interpretation of the results based on context are highlighted at the end of the discussion (page 18, lines 291-295):

“Since other infectious diseases may mimic their clinical presentation, namely bacterial sepsis, malaria, influenza, Covid-19 and rickettsial diseases, another limitation was the comparison of leptospirosis to DF only. Thus, our results should be interpreted with caution and not systematically transposed to areas/settings where such illnesses are largely prevalent.”

Comment:

The authors might reconsider expanding the study to include more and better-characterized samples, and perform a more sophisticated analysis including clinical and biochemical predictors and modelling to create a decision tree – this would result in an interesting publication.

Response:

The symptoms and tropism of leptospirosis, thus the determinants, are very much related to the Leptospira species involved. Basing one’s decision on these determinants is context-dependent and not very useful elsewhere because the distribution of Leptospira species varies between settings. However, CRP appears to be a common determinant, regardless of the species or serogroup involved, and thus could be extended to all settings.

Moreover, in our setting, CRP had a much better Wald test value than any other biological or clinical data, and a CRP threshold of 50 mg/L led to an AUC of 0.975, IC95% : 0.957-0.994 with a binary choice. In fact, this is confirmed by CART analysis which however does not seem to be crucial for concluding the performance and utility of CRP at a threshold of 50 mg/L to differentiate leptospirosis from dengue in clinical practice.

Finally, one of the limitations of decision trees is that they are largely unstable compared to other predicting models. A small change in the dataset can make the tree structure unstable which can cause variance. It is therefore recommended to balance the data set prior to fitting with the decision tree, as there is a risk of overfitting which can limit interpretation and generalization to other settings.

Major comment:

Analyses could have been expanded to include the construction of classification and regression trees, whereby using the additional clinic-laboratory findings to strengthen the predictions, versus CRP alone.

Response:

As discussed above, we believe that CART analysis was not crucial to conclude the performance and utility of CRP at a threshold of 50 mg/L to differentiate leptospirosis from dengue in clinical practice.

Minor comments:

All dengue cases were PCR-confirmed, but Leptospirosis cases included a serological positivity endpoint (Serion ELISA, IgM), which has its well-described limitations. In addition, no positivity cutoff was provided, and no justification for cut off selection.

Response:

Serology was not routinely done, but when there was a strong suspicion of dengue fever and a negative RT-PCR result, but this was infrequent because patients with dengue fever presented very early to emergency services whereas the incubation time for leptospirosis more often resulted in negative PCR results. 

And dengue serology was not systematically checked three weeks later during the epidemic, or in ambulatory care, so the link between the data could not be made with confidence. And a single positive IgM ELISA result might represent either an active or a previous, recent infection. Therefore, only dengue cases with positive RT-PCR results were included in the study. 

In contrast, leptospirosis cases are usually re-examined 1 month after diagnosis so a second IgM serology could be performed in the hospital to confirm the diagnosis of current leptospirosis. Therefore, leptospirosis cases diagnosed by serology were included.

Interpretation of results for the Serion ELISA classic Leptospira IgM was done according the manufacturer’s instructions : anti-leptospiral IgM <15 IU/ml gave a negative result suggesting no evidence of a recent infection, 15–19 IU/ml gave a borderline result suggesting that it may be a recent infection and ≥20 IU/ml gave a positive result which is interpreted as a recent or current infection to be compared with the result of a second IgM serology. Manufacturer’s protocol and performance of the Serion IgM ELISA kit have been reported elsewhere, for example :

- Dreyfus A, Ruf MT, Goris M, Poppert S, Mayer-Scholl A, Loosli N, Bier NS, Paris DH, Tshokey T, Stenos J, Rajaonarimirana E, Concha G, Orozco J, Colorado J, Aristizábal A, Dib JC, Kann S. Comparison of the Serion IgM ELISA and Microscopic Agglutination Test for diagnosis of Leptospira spp. infections in sera from different geographical origins and estimation of Leptospira seroprevalence in the Wiwa indigenous population from Colombia. PLoS Negl Trop Dis. 2022 Jun 6;16(6):e0009876. doi: 10.1371/journal.pntd.0009876. PMID: 35666764; PMCID: PMC9223614.

- Courdurie C, Le Govic Y, Bourhy P, Alexer D, Pailla K, Theodose R, Cesaire R, Rosine J, Hochedez P, Olive C. Evaluation of different serological assays for early diagnosis of leptospirosis in Martinique (French West Indies). PLoS Negl Trop Dis. 2017 Jun 23;11(6):e0005678. doi: 10.1371/journal.pntd.0005678. PMID: 28644889; PMCID: PMC5500375.

- Niloofa R, Fernando N, de Silva NL, Karunanayake L, Wickramasinghe H, Dikmadugoda N, Premawansa G, Wickramasinghe R, de Silva HJ, Premawansa S, Rajapakse S, Handunnetti S. Diagnosis of Leptospirosis: Comparison between Microscopic Agglutination Test, IgM-ELISA and IgM Rapid Immunochromatography Test. PLoS One. 2015 Jun 18;10(6):e0129236. doi: 10.1371/journal.pone.0129236. PMID: 26086800; PMCID: PMC4472754.

Of the 98 leptospirosis cases, 93 were confirmed by serum and/or urine PCR +/- IgM serology (see flowchart in Fig. 1) and 9 were confirmed by serology only.

We have therefore modified the text as follows (page 6, lines 103-109): “For Leptospira serology, the Serion ELISA classic IgM (Institut Virion\\Serion GmbH, Germany) was used. Interpretation of results was done according the manufacturer’s instructions : anti-leptospiral IgM <15 IU/ml gave a negative result suggesting no evidence of a recent infection, 15–19 IU/ml gave a borderline result suggesting that it may be a recent infection and ≥20 IU/ml gave a positive result which was interpreted as a recent or current infection to be compared with the result of a second IgM serology often performed in the hospital, because leptospirosis cases are usually re-examined 1 month after diagnosis.”

As only PCR results were considered to confirm dengue diagnosis in our study, we deleted the last sentence of the paragraph: "For DF screening, the Panbio dengue ELISA kit was employed.", otherwise it would be relevant to also describe the Panbio dengue ELISA, NS1 ELISA and NS1 RDT performed in our setting.

We hope you will find our revised version acceptable for publication in PLOS One and look forward to hearing from you.

Yours sincerely,

Dr Olivier MAILLARD, PharmD, MSPH

Dr Antoine BERTOLOTTI, MD, PhD

Dr Loic RAFFRAY, MD, PhD

---

## [Editor Report · Decision Letter 1]

4 May 2023

C-reactive protein: an easy marker for early differentiation between leptospirosis and dengue fever in endemic area

PONE-D-22-26652R1

Dear Dr. Maillard,

We’re pleased to inform you that your manuscript has been judged scientifically suitable for publication and will be formally accepted for publication once it meets all outstanding technical requirements.

Kind regards,

Andrea L. Conroy, PhD

Academic Editor

PLOS ONE
---

## [Editor Report · Acceptance letter]

8 May 2023

PONE-D-22-26652R1 

C-reactive protein: an easy marker for early differentiation between leptospirosis and dengue fever in endemic area 

Dear Dr. Maillard:

I'm pleased to inform you that your manuscript has been deemed suitable for publication in PLOS ONE. Congratulations! Your manuscript is now with our production department. 

Kind regards, 

on behalf of

Dr. Andrea L. Conroy 

Academic Editor

PLOS ONE